

# Benchmarking viromics: an *in silico* evaluation of metagenome-enabled estimates of viral community composition and diversity

Simon Roux[1], Joanne B. Emerson[1], Emiley A. Eloe-Fadrosh[2] and
Matthew B. Sullivan[1,3]

[1] Department of Microbiology, Ohio State University, Columbus, OH, United States of America
[2] Joint Genome Institute, Department of Energy, Walnut Creek, CA, United States of America
[3] Department of Civil, Environmental and Geodetic Engineering, Ohio State University, Columbus, OH, United States of America

Corresponding authors
Simon Roux, sroux@lbl.gov
Matthew B. Sullivan,
mbsulli@gmail.com

## ABSTRACT

**Background**. Viral metagenomics (viromics) is increasingly used to obtain unculti-vated viral genomes, evaluate community diversity, and assess ecological hypotheses. While viromic experimental methods are relatively mature and widely accepted by the research community, robust bioinformatics standards remain to be established. Here we used *in silico* mock viral communities to evaluate the viromic sequence-to-ecological-inference pipeline, including (i) read pre-processing and metagenome assembly, (ii) thresholds applied to estimate viral relative abundances based on read mapping to assembled contigs, and (iii) normalization methods applied to the matrix of viral relative abundances for alpha and beta diversity estimates.

**Results**. Tools specifically designed for metagenomes, specifically metaSPAdes, MEGAHIT, and IDBA-UD, were the most effective at assembling viromes. Read pre-processing, such as partitioning, had virtually no impact on assembly output, but may be useful when hardware is limited. Viral populations with 2–5× coverage typically assembled well, whereas lesser coverage led to fragmented assembly. Strain heterogeneity within populations hampered assembly, especially when strains were closely related (average nucleotide identity, or ANI ≥97%) and when the most abundant strain represented <50% of the population. Viral community composition assessments based on read recruitment were generally accurate when the following thresholds for detection were applied: (i) ≥10 kb contig lengths to define populations, (ii) coverage defined from reads mapping at ≥90% identity, and (iii) ≥75% of contig length with ≥1× coverage. Finally, although data are limited to the most abundant viruses in a community, alpha and beta diversity patterns were robustly estimated (±10%) when comparing samples of similar sequencing depth, but more divergent (up to 80%) when sequencing depth was uneven across the dataset. In the latter cases, the use of normalization methods specifically developed for metagenomes provided the best estimates.

**Conclusions**. These simulations provide benchmarks for selecting analysis cut-offs and establish that an optimized sample-to-ecological-inference viromics pipeline is robust for making ecological inferences from natural viral communities. Continued development to better accessing RNA, rare, and/or diverse viral populations and

improved reference viral genome availability will alleviate many of viromics remaining limitations.

# BACKGROUND

Microbial communities and their associated viruses are abundant, diverse, and play key roles in Earth's ecosystems and processes (*Falkowski, Fenchel & Delong, 2008*; *Cobián Güemes et al., 2016*). However, because most microbes and viruses remain uncultivated, and because viruses do not harbor a universal marker gene, viral ecology studies remain challenging to perform (*Brum & Sullivan, 2015*; *Solden, Lloyd & Wrighton, 2016*). Viral metagenomics (viromics) is a uniquely powerful tool for high-throughput analysis of uncultivated viruses (*Brum & Sullivan, 2015*; *Cobián Güemes et al., 2016*). Initial viromics studies, despite being limited to gene-level analyses, revealed the large diversity of viral-encoded genes (*Edwards & Rohwer, 2005*; *Schoenfeld et al., 2008*), provided first estimates of richness and functional diversity across natural viral communities (*Hurwitz, Hallam & Sullivan, 2013*; *Hurwitz, Brum & Sullivan, 2015*), and suggested the existence of biome-specific viral communities distributed worldwide (*Rodriguez-Brito et al., 2010*; *Roux et al., 2012*).

Thanks to recent improvements in high-throughput sequencing technologies and genome assembly, viromes now also provide the opportunity to assemble large genomes fragments (and even complete genomes) of uncultivated viruses (reviewed in *Brum & Sullivan, 2015*; *Rose et al., 2016*). Historically, *in silico* benchmarks of the assembly process for microbial metagenomes indicated that accurate bacterial and archaeal genomes (complete or partial) could be recovered for relatively abundant lineages given sufficient sequencing depth, but revealed potential issues including misassemblies deriving from the presence of very closely related organisms (*Mavromatis et al., 2007*; *Mende et al., 2012*; *Greenwald et al., 2017*; *Sczyrba et al., 2017*). Viral community datasets are typically processed using the same methodologies, and viral-specific benchmarks came to a similar conclusion: viral genomes can be assembled from metagenomes, but the presence of co-existing viruses with highly similar regions in their genome can lead to reduced contig size and/or chimeric contigs (*Aguirre de Cárcer, Angly & Alcamí, 2014*; *Vázquez-Castellanos et al., 2014*; *García-López, Vázquez-Castellanos & Moya, 2015*; *Martinez-Hernandez et al., 2017*; *White, Wang & Hall, 2017*). However, new metagenome assembly softwares (e.g., metaSPAdes, *Nurk et al., 2017*) and methods for read filtering and/or partitioning prior to assembly (e.g., khmer, *Crusoe et al., 2015*) that might improve assembly quality have yet to be evaluated with viral data.

For bacteria and archaea, advances in genome binning and genome validation approaches (e.g., *Parks et al., 2015*) have significantly improved the recovery of accurately reconstructed genomes from increasingly complex environments (*Wrighton et al., 2012*; *Sharon et al., 2013*; *Waldor et al., 2015*; *Sangwan, Xia & Gilbert, 2016*; *Sczyrba et al., 2017*).

These methods rely on single-copy marker genes to assess genome bin completeness and "contamination" (i.e., multiple genomes in the same genome bin), two metrics critical to guide the optimization of genome binning parameters and curate the final dataset (*Parks et al., 2015*; *Bowers et al., 2017*). Unfortunately, because of the absence of universal single-copy viral marker gene, viral genome bins are much more challenging to interpret and analyze. Since viral genomes are also smaller than microbial ones and thus more frequently assembled in a single contig, viromics studies usually rely on the assembled contigs without applying any genome binning step.

For ecological analyses, a community abundance matrix of microbial OTU counts across samples is the typical starting point, and this "OTU table" is often derived from 16S rRNA gene abundances in amplicon sequencing datasets or metagenomes (*Hill et al., 2003*; *Roesch et al., 2007*; *Fulthorpe et al., 2008*; *Fierer et al., 2011*; *Logares et al., 2014*). Even for these relatively established microbial ecological analyses, appropriate normalization methods that account for different sequencing throughput across samples are still debated, and rarely are results compared across multiple normalization methods to establish best practices (*Doll et al., 2013*; *Paulson et al., 2013*; *McMurdie & Holmes, 2014*). This microbial ecology pipeline also needs adjustment when applied to viruses because viruses lack a universal marker gene, precluding amplicon-based viral population abundance estimates at the community scale (although amplicon-based studies have been successful for ecological analyses of specific viral lineages, e.g., *Filée et al., 2005*; *Goldsmith et al., 2011*; *Chow & Fuhrman, 2012*). Notably, comparative genomic and ecological analysis of model systems enabled the identification of sequence-discrete populations, which represent stable ecotypes in natural viral communities (*Marston & Amrich, 2009*; *Gregory et al., 2016*; *Marston & Martiny, 2016*). Thus, in the absence of a universal viral marker gene, these genome-based populations have been proposed to be used as a viral population units (akin to a microbial operational taxonomic unit, OTU) in ecological analysis (*Brum et al., 2015*). Pragmatically, viral populations are derived from *de novo* metagenomic assemblies, with abundances estimated by metagenomic read recruitment. Ecological analyses of these contig-derived abundance matrices still have to be comprehensively evaluated, although one bias specific to this approach has already been identified: counting each assembled contig as a separate OTU can lead to over-estimates of the number of different viruses in the community (*Aziz et al., 2015*; *García-López, Vázquez-Castellanos & Moya, 2015*).

Here we used 14 *in silico* simulated viral metagenomes to (i) compare the assembly results across different reads pre-processing methods and assemblers, both in terms of the overall genomes recovery and the number and type of errors observed, (ii) assess potential biases and identify optimal thresholds for identification and quantification of viral populations from metagenomic contigs, and (iii) determine if virome populations abundance matrices can provide reliable estimates of alpha diversity (i.e., diversity within a community) and beta diversity (i.e., differentiation between communities), even in cases where sequencing depth vary widely (up to two orders of magnitude) between samples.

## METHODS

### Mock community design

Viral genomes were randomly selected among the complete genomes of viruses infecting bacteria or archaea in the NCBI RefSeq database (v69, 2015-02). For each mock community, the total number of viruses randomly selected (between 500 and 1,000, Table S1, Fig. S1A), as well as the parameter of the power law distribution used to model relative abundances (between 1 and 50) were varied (Figs. S1B–S1D). To create patterns of beta diversity across samples, the 50 most abundant viruses were homogenized within each of four sample groups, i.e., samples within a group shared 30 to 50 of their most abundant viruses, and samples between groups did not share any of their most abundant 50 viruses. This led to a clear beta diversity pattern with the mock communities clustering into four groups (Figs. S1E & S1F, a PerMANOVA was performed in R with the package vegan (Oksanen et al., 2017) to verify that the sample groups were significantly different).

### Virome simulations

To simulate virome sequencing for each mock community, the number of reads derived from each genome was first calculated based on the relative abundance of the genome in the mock community and the total number of reads sequenced in the virome (10 millions paired-end reads in the initial viromes, 1 million and 100,000 paired-end reads for the subsets at 10% and 1% respectively). Then, NeSSM (Jia et al., 2013) was used to generate random reads ($2 \times 100$ bp) at the prescribed abundances with simulated Illumina HiSeq errors.

### Reads processing

Reads generated by NeSSM were first quality-controlled with Trimmomatic (Bolger, Lohse & Usadel, 2014) with a minimum base quality threshold of 30 evaluated on sliding windows of 4 bases, and minimum read length of 50. We opted not to evaluate different error correction softwares or to compare raw reads to quality-controlled (QC) reads, as previous studies have already provided such benchmarks for genomic assembly, which should be applicable to metagenomic assembly as well (e.g., Yang, Chockalingam & Aluru, 2013).

All sets of additionally pre-processed reads were generated from these QC reads using khmer v1.4.1 (Crusoe et al., 2015), following the online protocols (http://khmer-protocols.readthedocs.io/, Fig. S2). First, a dataset of digitally normalized reads was generated, i.e., a dataset in which all reads with median k-mer abundance higher than a specified threshold were eliminated. This was done in two steps by normalizing k-mer coverage first to $20\times$ then to $5\times$ (script "normalize-by-median", dataset "Digital normalization"). The script "do-partition" was then used to partition these digitally normalized datasets, i.e., separate reads that did not connect to each other in the k-mer graphs in different bins (dataset "Partitioned reads (normalized)"). These reads partitions were then re-inflated, i.e., the original abundance of reads was restored to its value prior to digital normalization, with the script "sweep-reads" (dataset "Partitioned reads (inflated)"). Finally, three sets of reads were generated by trimming all low-abundance k-mers for highly covered reads, i.e., highly covered reads (in this case, $\geq 20\times$) were

truncated at the first occurrence of a k-mer below a given abundance cutoff (here $\leq 2\times$, $\leq 5\times$, and $\leq 20\times$ for the three datasets "Low k-mer filter $(2\times)$", "Low k-mer filter $(5\times)$", "Low k-mer filter $(20\times)$", respectively). This was done with the script "filter-abund", with option "variable-coverage" as recommended for metagenomes.

## Assembly and comparison to input genomes

The different read sets were assembled with five different assembly software tools, using metagenomic-optimized parameters (when available, Fig. S2). IDBA-UD v.1.1.1 (*Peng et al., 2012*) was used with the option "pre-correction" and from fasta reads (converted from fastq reads with the tool "fq2fa"). MetaSPAdes assemblies (*Nurk et al., 2017*) were computed from the software version 3.10.0, with the option "metagenomic" (all other options default). MEGAHIT assemblies (*Li et al., 2016*) were computed from version v1.0.6 with presets "meta" (all other options default). MetaVelvet assemblies (*Namiki et al., 2012*) were computed with software version 1.2.07 with the "discard_chimera" option selected, default parameters otherwise. Omega assemblies (*Haider et al., 2014*) were computed with software version 1.0.2 and minimum overlap length of 60. Each assembler was applied to each read pool from each sample (7 read pools $\times$ 14 samples = 98 assemblies, Fig. S2), retaining all contigs $\geq$500 bp for each assembly (Table S4).

Contigs were compared to the input genomes with nucmer (*Delcher, Salzberg & Phillippy, 2003*)(default options). When $\geq$95% of a contig's length matched an input genome at $\geq$90% nucleotide identity, that contig was considered to be a genuine assembly of the input genome. Otherwise, if a contig was similar to multiple genomes but to none over $\geq$95% of its length, it was considered a chimera. Circular contigs were detected based on identical 5′ and 3′ ends, as in (*Roux et al., 2014*). A circular contig with a length corresponding to $\geq$95% of the original genome length was considered a genuine complete genome assembly, while circular contigs covering less than 95% of the original genomes were considered false positives (i.e., incomplete contigs incorrectly predicted as complete genome assemblies). R was used to conduct $t$-test when comparing rate of chimeric contigs across assemblers and reads pre-processing methods, using the assembly of QC reads with MEGAHIT as the control (the set of contigs with the lower number of chimeras).

## Generation of the non-redundant pool of population contigs and coverage estimation

Based on the previous benchmarks, the assemblies obtained with metaSPAdes from the QC reads were considered to be the most optimal assemblies and were used in all subsequent benchmarking analyses. Contigs from all samples were clustered with nucmer (*Delcher, Salzberg & Phillippy, 2003*) at $\geq$95% ANI across $\geq$80% of their lengths, as in (*Brum et al., 2015*; *Gregory et al., 2016*), to generate a pool of non-redundant "population contigs". QC reads from each sample were then mapped to these population contigs with bbmap (http://bit.ly/bbMap), with ambiguous mapping assigned to contigs at random (option ambiguous=random). A custom python script was then used to estimate the number of reads and coverage of each contig.

## Alpha and beta diversity estimates

The abundance of each population contig in a given sample was estimated based on the number of reads mapping to that contig, normalized by the contig length (to account for differences in contig / genome size). Beyond the raw read counts (normalized by contig length), five abundance matrices were generated with different library size normalization methods as follow (summarized in Fig. S2):

- "Normalized": counts were divided by the total library size, i.e., the total number of QC reads in the sample, as used for example in *Brum et al. (2015)*. This approach is also known as "total-sum scaling".
- "MGSeq": counts were normalized through cumulative-sum scaling with the metagenomeSeq R package (*Paulson et al., 2013*). This method was specifically designed for metagenomes in which communities are under-sampled (as is the case in most viral metagenome studies), and will divide counts by a cumulative sum of count to a given percentile (as opposed to dividing by total counts as in "Normalized"). This will minimize the effects of the few highly abundant viruses potentially dominating the community, and introducing biases in relative abundances (*Paulson et al., 2013*).
- "EdgeR": counts were normalized using scaling factors for libraries designed to minimize the log-fold change between samples for most of the populations, computed with the edgeR R package (*Robinson, McCarthy & Smyth, 2009*). This method was initially developed for count-based expression data and assumes that the relative abundances of most features (here populations) will not vary between two samples.
- "DeSeq": as with EdgeR, counts were normalized to minimize variations between samples for most populations but with a different underlying model, computed with the DESeq R package (*Anders & Huber, 2010*). As with EdgeR, this method was initially developed for the detection of differentially expressed features in sequence count data analysis.
- "Rarefied": new counts were generated based on rarefied sets of reads, i.e., quality-controlled reads are subsampled (without replacement) to the smallest number of quality-controlled reads across all samples. Thus, all of the libraries are artificially set to the same size, however some data are "wasted" in the process, i.e., for the more deeply sequenced samples, some observations will not be included in the rarefied counts (*McMurdie & Holmes, 2014*).

Each abundance matrix was then used to calculate alpha and beta diversity indices, namely the Shannon index, Simpson index, and pairwise Bray–Curtis dissimilarities between samples with a custom perl script. R was used to generate all plots using the ggplot2 package (*Wickham, 2009*), as well as the NMDS and PerMANOVA analyses, computed with the vegan package (*Oksanen et al., 2017*). For alpha diversity, we opted to only test indices reflecting community structure (Shannon and Simpson indexes) and not indices predicting sample richness (e.g., Chao estimators (*Chao, 1984*)), since the latter have been highlighted as not suitable for cases in which rare members of the community are not adequately sampled (*Haegeman et al., 2013*).

### Under-sequencing and strain heterogeneity benchmarks

To evaluate the impact of under-sequencing on alpha and beta diversity estimates, the same pipeline (assembly with metaSPAdes from QC Reads, selection of population contigs, and estimation of alpha and beta diversity) was applied to datasets in which seven of the 14 samples were under-sequenced. Two levels of under-sequencing were tested, one in which under-sequenced samples were set at 10% of the initial library size (i.e., 1,000,000 reads) and another at 1% of the initial library size (100,000 reads, Table S1).

To evaluate the impact of strain heterogeneity (within-population genomic diversity) on assembly success, a custom perl script was used to simulate strain variations as observed on natural populations of T4-like cyanophages (*Gregory et al., 2016*), i.e., a set of potentially mutated positions were determined for each new simulated strain gathering all intergenic positions, all third codons positions in protein-coding genes, and all positions in two randomly selected genes (to simulate genes undergoing diversifying selections). These simulations were based on the mock community "Sample_1", for which every genome was transformed into a population composed of a set of related strains.

For each population, three parameters selected randomly and independently:

- The total number of strains was set at 10, 50, or 100 strains simulated.
- The strain divergence, controlled by a "mutation rate", i.e., the ratios of positions mutated within the set of positions identified as "potentially mutated" (see above). The other positions in the genome, not selected as potentially mutated, were mutated at a rate 100 times lower. This "mutation rate" was set at 5%, 10%, or 20%. This led to ANI between the generated strains and the original reference genomes of 97–100%, 95–97%, and 90–95%, respectively.
- The relative abundance of individual strains within the population, sampled from a power-law distribution. The shape of the distribution was controlled by the power-law parameter, set at 0.1, 1, 10, 100, or 1,000. This led to the dominant (i.e., most abundant) strain representing from 1% to 100% of the population.

For each population, reads were then simulated with NeSSM (*Jia et al., 2013*), with the total reads generated for each population calculated based on the input coverage (as for previous simulations), and the number of reads generated from each strain calculated from the strains relative abundance. Reads were then processed as previously, i.e., quality-controlled, partitioned, or filtered, and assembled with the five assemblers tested using the same options as for the simulated viromes. Finally, the size of the largest contig recovered for each population was compared to the size of the largest contig recovered for the same genome without strain heterogeneity, to evaluate the impact of strain heterogeneity independently from differences in assembly efficiency between coverage levels, reads processing methods, and assemblers.

## RESULTS AND DISCUSSION

### Mock communities design

A set of 14 viral communities was designed to provide a gradient of alpha diversity and clear beta diversity patterns (Fig. S1, Tables S1 & S2). These communities were composed

of 500 to 1,000 genomes (randomly sampled within bacteriophages and archaeal viruses available in NCBI RefSeq v69), with the relative abundance of individual genomes based on power law distributions with varying exponents. These simulations are thus designed to reflect a diverse viral community, as is usually observed in environmental samples (e.g., oceans, lakes, soils, or human gut), but would not correspond to viral communities dominated by a single type of virus, e.g., clinical samples associated with a specific host or epidemiological samples targeting a specific type of virus. Beyond differences in alpha diversity, these communities were also designed to organize into four "ecological" clusters, i.e., four groups of mock communities sharing more genomes within than between groups (Fig. S1). Thus, this simulated dataset allows us to evaluate the ability of virome-based population ecology approaches to recover absolute values of alpha diversity, as well as trends in alpha diversity and beta diversity patterns across samples.

Virome reads were simulated *in silico* with NeSSM (*Jia et al., 2013*) for each mock community (10,000,000 paired-end Illumina HiSeq reads, $2 \times 100$ bp). Since the number of reads derived from each genome was based on its prescribed relative abundance in the community, 29.1% to 75.2% of the viral genomes in each mock community did not get "sequenced" at all (i.e., did not yield any reads). This was by design to mimic the lack of sampling for rare viruses by current sequencing efforts of environmental samples.

### Testing the capacity and accuracy of assembly tools

Given metagenomic sequence data from these 14 mock communities, we first evaluated currently available assembly algorithms. To this end, five assemblers (IDBA-UD (*Peng et al., 2012*), MEGAHIT (*Li et al., 2016*), MetaVelvet (*Namiki et al., 2012*), Omega (*Haider et al., 2014*), and metaSPAdes (*Nurk et al., 2017*), all adapted to assemble metagenomic data) were compared to assess their ability to accurately assemble genomes of bacterial and archaeal viruses from viromes (Fig. S2). As expected, each of the assemblers successfully assembled highly covered genomes ($10\times$ or higher) and failed to assemble most low-coverage genomes ($2\times$ and lower, Fig. 1A, Fig. S3A). However, MetaVelvet and Omega required higher coverage to assemble viral genomes ($\sim 5-10\times$), while IDBA-UD, MEGAHIT, and metaSPAdes routinely assembled genomes at $\sim 2-5\times$ coverage (Fig. 1A, Fig. S3A). A similar trend was found when observing genome recovery in a single contig (i.e., the percentage of a genome assembled in a single contig, as opposed to the percentage of a genome assembled when cumulating all contigs). Again, IDBA-UD, MEGAHIT, and metaSPAdes were more efficient than MetaVelvet and Omega for assembling large genome fragments at lower read coverage ($\sim 2-20\times$), and metaSPAdes was also better than IDBA-UD and MEGAHIT for assembling low-coverage genomes in a single large contig (Fig. 1B, Fig. S3B).

When comparing individual genome assemblies across the three best assemblers (metaSPAdes, IDBA-UD, and MEGAHIT), no clear differences could be observed in the genome recovery (Fig. S4, correlation coefficients between assemblers > 0.99). However, the percentage of each genome recovered in a single contig was more variable among assemblers (Fig. S4, correlations coefficients: 0.88–0.98). This comparison did not indicate that one assembler would be systematically better than another, but rather that the best assembly for a given genome could come from any of these three assemblers.

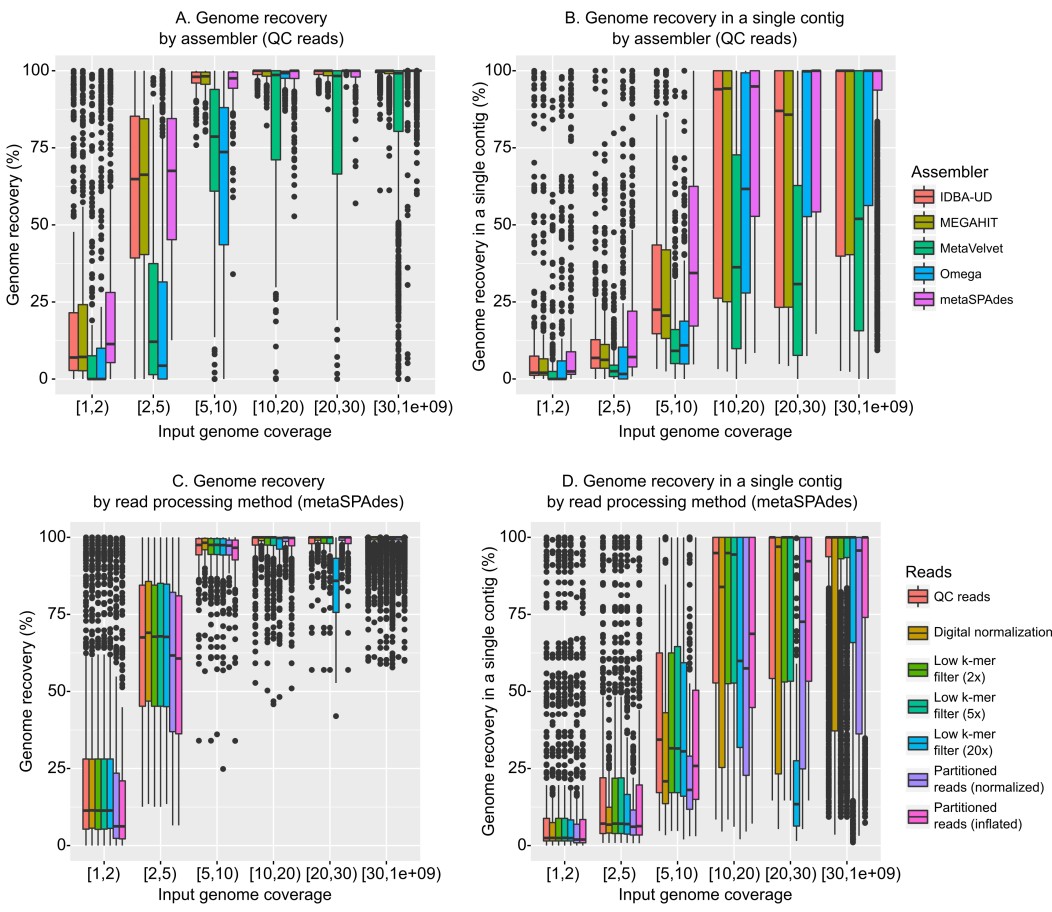

**Figure 1** **Influence of assembly software and read curation on genome recovery.** All plots display the input coverage on the *x*-axis, and either the cumulated genome recovery across all contigs (A & C) or the highest genome recovery by a single contig (B & D) on the *y*-axis. (A & B) display a comparison of assemblers applied to quality-controlled (QC) reads. (C & D) present a comparison of read pre-processing methods, all assembled with metaSPAdes. Comparable plots for reads assembled with the other assemblers are available in Fig. S5.

Together these comparisons suggest that: (i) IDBA-UD, MEGAHIT, and metaSPAdes are currently the best available choices for maximizing assembly of viral contigs from short-read (100 bp) viromes (assembly accuracy discussed below), (ii) regardless of the choice of assembly tool, low coverage genomes ($<2\times$) are under-assembled, and (iii) because assembly success varies across genomes and assemblers, multiple tools should be compared to optimally assemble desired target genomes from viromes. Overall, these results are consistent with microbial metagenomic benchmarks, which also indicated that assemblers designed specifically for metagenomes, especially metaSPAdes, MEGAHIT, and IDBA-UD, provided the best assemblies (*Sczyrba et al., 2017*; *Vollmers, Wiegand & Kaster, 2017*).

## Impact of k-mer-based read filtering and partitioning on assembly

Next, we evaluated how available read pre-processing approaches impacted genome assembly (using approaches from the khmer package and summarized in Table S3 and Fig. S2) (*Crusoe et al., 2015*). Briefly, beyond the reference dataset of quality controlled reads, the

different methods tested were (i) trimming of reads based on low-abundance k-mers, i.e., reads are truncated at the first occurrence of a low-abundance k-mer likely originating from sequencing error, (ii) digital normalization, i.e., the removal of redundant sequences to normalize genome coverage at or under a specific value (here 5×), and (iii) read partitioning, i.e., separate assembly of the disconnected components of the k-mer graph.

Overall, and compared with the effect of the different assembly algorithms, the read pre-processing had a minimal impact on the assembly output (Figs. 1C and 1D, Figs. S3C & S3D with metaSPAdes; the same observations were made with different assemblers in Fig. S5). The main effects observed were that (i) digital normalization (treatments "Digital normalization" and "Partitioned reads (normalized)") led to sub-optimal assemblies, likely because differences in coverage above 5× are useful for assemblers to distinguish between related genomes, and (ii) trimming of low-abundance k-mers led to sub-optimal assemblies when the threshold used to define low abundance k-mers was close to the threshold used to define "abundant" reads to be trimmed (effect especially noticeable for the 20× filter, Figs. 1C & 1D). Conversely, partitioning reads and keeping their coverage information (treatment "Partitioned reads (inflated)") or trimming low-abundance k-mers from high coverage reads (with thresholds of 2× and 5×) had little effect on the assembly output, except on low-coverage genomes (<5×). These observations are consistent with the initial expectations of khmer's performance (*Crusoe et al., 2015*), although these simulations illustrate that digital normalization alone (i.e., without read partitioning and restoration of original read coverage) can lead to a sub-optimal metagenomic assembly.

## Errors and limitations of genome assembly from viromes

Beyond the assembly of low-coverage genomes, which was found to be challenging for all assemblers tested, other errors are known to occur during the *de novo* assembly of viromes.

First, chimeric contigs (i.e., contigs representing artificial constructs assembled from two or more distinct genomes) were generated in each assembly, as previously noted (*Aguirre de Cárcer, Angly & Alcamí, 2014*; *Vázquez-Castellanos et al., 2014*; *García-López, Vázquez-Castellanos & Moya, 2015*). In our simulated data, these usually represented less than 2.5% of the assembled datasets, and less than 5% of the large contigs (≥10 kb), but these numbers varied between assemblers and read curation methods (Figs. 2A & 2B). This low number of chimeric contigs is in accordance with benchmarks of microbial metagenomes, and suggests that metagenome assemblers in general can correctly reconstruct microbial and/or viral genomes (*Mende et al., 2012*). For all assemblers, reads after digital normalization always yielded more chimeric contigs, which confirmed that the digital normalization step led to sub-optimal assemblies ($p$-value <0.01). MEGAHIT systematically produced fewer chimeric contigs than IDBA-UD and metaSPAdes, especially for large (≥10 kb) contigs (Fig. 2B, $p$-value < 0.01). Hence, although MEGAHIT did not assemble as many large genome fragments, the fragments that were assembled contained fewer chimeras.

Next, we investigated whether finished and closed viral genomes assemblies could be robustly identified as "circular" contigs, i.e., contigs with matching 5′ and 3′ ends, as previously suggested (*Roux et al., 2014*). The ratio of false-positive circular contigs, i.e., circular contigs that represented less than 95% of the original genome and thus likely arose from

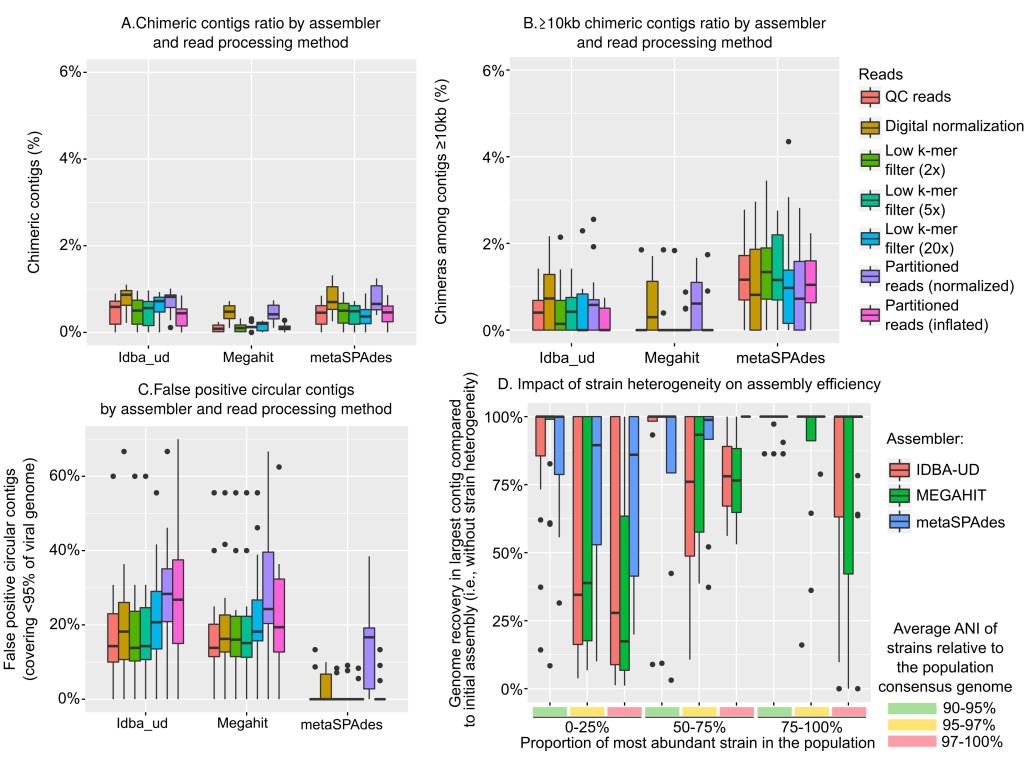

**Figure 2** **Types and frequency of errors observed in genome assembly from viral metagenomes.** (A) Percentage of chimeric contigs (i.e., contigs originating from two distinct genomes) across all assembled sequences, by assembler ($x$-axis) and read curation method (colors). (B) Percentage of chimeric contigs among large ($\geq$10 kb) contigs, by assembler ($x$-axis) and read curation method (colors). (C) Percentage of false-positive circular contigs, i.e., contigs identified as circular (matching 5′ and 3′ ends) but representing 95% or less of the original genome, by assembler ($x$-axis) and read curation method (color). (D) Impact of strain heterogeneity (i.e., presence of multiple strains from the same population) on the assembly efficiency. These tests were computed on one mock community (Sample_1), for which each reference genome was replaced with a set of related strains with varying divergence and relative abundances. The $y$-axis represents the ratio between the largest contig assembled for a genome when strain heterogeneity is introduced and the same parameter without strain heterogeneity (i.e., previous assemblies of the same Sample_1). Populations are grouped based on the two main parameters explaining assembly inefficiency: proportion of the most abundant strain in the population (C, D) and divergence of strains in the population (A, B). Data presented here include assemblies from QC reads with IDBA-UD, MEGAHIT, and metaSPAdes, while the full set of parameters and approaches tested are presented in Fig. S6.

repeat regions within a genome, was not modified by read pre-processing but was different among assemblers (Fig. 2C). Specifically, 10 to 30% of the circular contigs generated by MEGAHIT and IDBA-UD did not correspond to a complete genome, while metaSPAdes assemblies rarely included any false positive (4 contigs, or <2%, for metaSPAdes assemblies of quality-controlled reads). This suggests that metaSPAdes circular contigs are more likely to correspond to complete genomes and that the "circularization" of a contig cannot be considered as proof of completeness for MEGAHIT and IDBA-UD contigs.

Finally, we evaluated the impact of population strain heterogeneity— i.e., the co-existence of closely related strains with distinct genomes from the same population—on virome assembly. In microbial communities, strain heterogeneity is known to considerably

hamper the assembly of the corresponding genomes (*Sharon et al., 2015*; *Martinez-Hernandez et al., 2017*; *Sczyrba et al., 2017*). Population genetic studies of natural viral communities are however challenged by the paucity of cultivated systems that include multiple viral genomic representatives from a single population. Pragmatically, this means that although strain heterogeneity has been observed for specific model systems (*Gregory et al., 2016*; *Marston & Martiny, 2016*), community-wide strain variations that would accurately reflect natural viral communities cannot be pulled from these data. Hence, we opted to generate a mock community using the same populations and relative abundances as Sample 1 above, but introduced some level of strain heterogeneity for each population by varying a combination of three parameters: (i) the number of strains in the population, either low ($n = 10$), medium ($n = 50$), or high ($n = 100$), (ii) the diversity of these strains, presented as the average ANI of strains compared to the consensus population genome, either low (90–95%), medium (95–97%), or high (97–100%), and (iii) the evenness of the power-law distribution of strain frequency in the population, either low (dominant variant represents 75–100% of the population), medium (dominant variant 50–75%), or high (dominant variant < 25%). For each genome, reads were thus not generated from the reference genome sequence as before, but from a set of strains generated and sampled using a random combination of these 3 parameters. Then, the same pipeline of read processing and assembly was applied, and the size of the largest contig obtained for each population was compared to the size of the largest contig obtained in the previous mock community assembly (i.e., without strain heterogeneity, Fig. 2D and Fig. S6).

An ANOVA was performed on the complete dataset (i.e., all combinations of assemblers and read processing) to evaluate which component of strain heterogeneity impacted the assembly process (see 'Methods'). The three parameters (number of strains, strain diversity, and evenness of strain distribution) significantly but differently impacted the assembly: population shape (i.e., strain distribution) was the main explanatory variable of suboptimal assemblies (*F*-value 149.8, *p*-value < 1*e*−16), strain diversity was also a strong driver of assembly failures (*F*-value 70.4, *p*-value < 1*e*−16), while the number of strains in the populations had a more marginal effect (*F*-value 2.8, *p*-value 0.06). Overall, when compared to the assemblies generated without strain heterogeneity, contigs were shorter for populations with an even strain distribution (i.e., dominant strain ≤ 50% of the population) and/or when strains were more similar to the consensus genome (i.e., average ANI to consensus ≥ 97%) and to each other, with the combination of both leading to the greater reduction in contig length (Fig. 2D). These results indicate that strain heterogeneity within natural viral populations will likely be a key factor contributing to assembly success and failure, and populations of evenly distributed closely related strains will be the most likely to fail to assemble in virome studies. A similar trend was observed for microbial genomes in the Critical Assessment of Metagenome Interpretation benchmarks, where the assembly of closely related genomes (i.e., those with strain-level heterogeneity) was found to be challenging for all assemblers tested, although the experimental design did not allow the evaluation of which level and parameter of strain heterogeneity were most impactful (*Sczyrba et al., 2017*).

## Population identification and quantification

In viral ecological studies, the next step after assembly often consists of identifying viral populations (i.e., contigs representing individual populations) and quantifying their relative abundances in each sample. We opted to use the contigs assembled with metaSPAdes from quality-controlled reads, as they represented the largest contigs overall across the different samples (despite ∼1% chimerism). We pooled contigs generated from all samples into a single non-redundant database (contigs were clustered at ≥95% of nucleotide identity across ≥80% of the contig length, in accordance with population genome analysis (*Gregory et al., 2016*)). Quality-controlled reads were then mapped to this database to estimate contig coverage across the 14 samples. Two types of thresholds were evaluated in this mapping step: (i) minimum nucleotide identity for a given read to be considered mapped to a given contig, and (ii) minimum length of the contig covered to consider a contig as "detected" in a sample (Fig. S2). Reads not meeting the threshold were removed from abundance counts, and contigs not meeting the detection threshold in a given sample were given abundance values of zero for that sample in the resulting coverage table.

Considering all non-redundant contigs ≥500 bp as different populations, we observed that increasing the two thresholds (read mapping identity percentage and length of contig covered) progressively decreased the sensitivity of the analysis (evaluated here as the percentage of genomes recovered among genomes which were covered ≥1× in the sample, Fig. 3A) and the false discovery rate (or FDR, which is the percentage of contigs recovered that were not part of the initial community, i.e., these genomes did not provide any reads to the simulated metagenome, Fig. 3B). However, because FDR decreased more precipitously than sensitivity, there is an optimal combination of thresholds for which FDR can be minimized and sensitivity maximized. In these simulations, that optimal threshold was ≥75% on the contig length coverage associated with ≥90% nucleotide identity for the read mapping, which led to a 3% decrease in sensitivity (compared to the most permissive thresholds), but only 13% FDR (compared to 49% for the most permissive thresholds).

As noted by previous studies (*Aziz et al., 2015*; *García-López, Vázquez-Castellanos & Moya, 2015*), considering all non-redundant contigs as distinct populations strongly over-estimated the total number of populations (on average, two to three contigs were counted for each individual genome, Fig. 3C). Thus, we re-analyzed our dataset using only non-redundant contigs ≥10 kb or circular as was proposed previously, and as required for taxonomic classification by gene content network-based analysis (*Bolduc et al., 2017*). Again, the optimal threshold combination was ≥75% of the contig length covered and ≥90% read mapping identity (Figs. 3D–3F). However, while sensitivity declined slightly (∼15%) compared to the dataset including all contigs ≥500 bp, FDR improved drastically to 0.2%, compared to 13% observed in the above analyses. Further, by increasing the stringency of the population definition, the number of contigs per genome that were counted as a population was 1.2 which is much closer to the correct number of 1 contig per genomne. More generally, increasing this contig size threshold quickly decreased the number of contig observed per genome, and most of the over-estimation observed earlier seemed to arise from contigs <5 kb (Fig. S7).

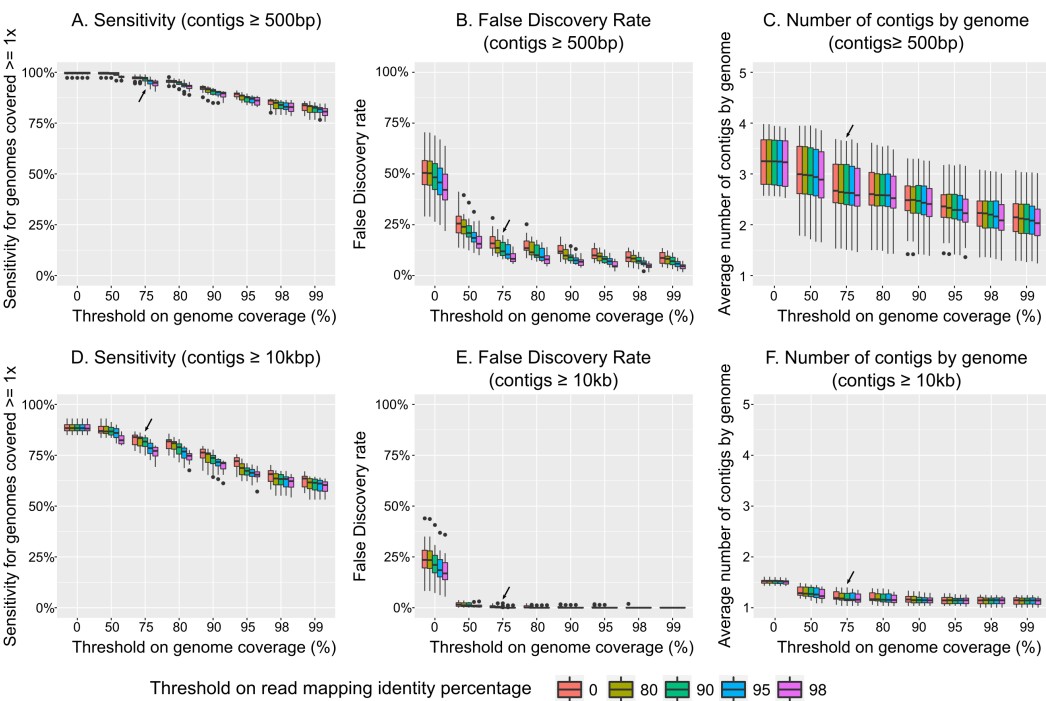

**Figure 3  Impact of read mapping thresholds on accuracy of viral population detection.** Two parameters were investigated when parsing the mapping of individual virome reads to the population contigs pool: (i) the percentage of a contig covered by a sample to considered the contig as detected (*x*-axis), and (ii) the percentage of identity of reads mapping to the contig (color scale). Two pools of population contigs were tested: all non-redundant contigs of ≥500 bp (A–C), and all non-redundant contigs ≥10 kb (D–F). Three metrics were calculated to evaluate the impact of mapping reads thresholds. The detection sensitivity is estimated as the percentage of "expected" genomes (i.e., genomes covered ≥ 1× in the sample) that were detected through mapping to population contigs (A and D). The false-discovery rate corresponds to the percentage of contigs detected in a sample through mapping to population contigs, but were not associated with any genomes from the initial sample (i.e., these genomes did not provide any reads to the simulated virome, so these contigs should not be detected, B and E). Finally the average number of distinct population contigs detected is calculated for each individual genome initially covered ≥1×, and correspond to the number of times a single genome is "counted" (i.e., multiple contigs suggest multiple populations, even though it is really just one population, C and F).

In summary, we recommend that viral populations (as an operational taxonomic unit) be defined and analyzed in viromes using contigs that are ≥10 kb or circular, and only considered "detected" when the contig is covered over ≥75% of its length by read mapping at ≥90% nucleotide identity. However, we also anticipate that the data from these sensitivity analyses will help researchers tune these thresholds to match a given study's need for high sensitivity or low FDR. Importantly though, these suggestions are specific to viromes, since microbial metagenomic studies can rely on genome binning and universally conserved, single-copy marker genes to estimate more robustly the global number and completeness of the different genomes assembled (*Sczyrba et al., 2017*).

## Alpha and beta diversity estimation from virome-derived populations
We next sought to evaluate how the variation in community structure of our 14 mock community metagenomes impacted diversity estimations, and did so using our

recommended optimized population cut-offs for identifying populations and then estimating their abundances by read mapping. These population count matrices (counting either base pairs or reads mapped to each population contig) were used as input for alpha and beta diversity estimations and compared across the dataset. Notably, these matrices included only a fraction (10–33%) of the original genomes in the dataset, as rare viral genomes were not "sequenced", and low-coverage genomes produced only small (<10 kb) contigs (Fig. 4A).

Before calculating any index, the read counts were first normalized by the contig length, since viral genome lengths can be highly variable (∼2 orders of magnitude, *Angly et al., 2009*). Then, to account for potential differences in library sizes, we compared five different methods: (i) a simple normalization in which counts are divided by the library size, "Normalized" (ii) a method specifically designed to account for under-sampling of metagenomes, from the metagenomeSeq R package, "MGSeq" (iii and iv) two methods designed to minimize log-fold changes between samples for most of the populations, from the edgeR R package, "edgeR", and the DESeq R package, "DESeq", and (v) a rarefaction approach whereby all libraries get randomly down-sampled without replacement to the size of the smallest library, "Rarefied" (Fig. S2).

For both Shannon and Simpson alpha diversity indices, the values calculated from normalized count matrices were within 10% of the actual value calculated from the whole community (Figs. 4B & 4C). Hence, the recovery of abundant members of the community seems to be enough to estimate alpha diversity values. Since both Shannon and Simpson indices are based on the relative abundance of individual members of the community, the three methods that applied a sample-wide correction factor (normalization by library size, MGSeq, EdgeR) all led to the same estimations, while rarefied count matrices and DESeq, which can (slightly) modify relative abundance of populations within communities, provided statistically indistinguishable estimates (Figs. 4B & 4C). Similarly, for beta diversity estimates, pairwise Bray–Curtis dissimilarities between samples calculated from normalized counts matrices were highly similar to the dissimilarities calculated from the whole communities for all normalization methods (within 15% of actual values, $p$-value ≤0.001 for Mantel test comparing true and estimated dissimilarity matrices, Fig. 4D). Thus, as long as the count matrices were normalized to account for different contig lengths and library sizes, each of the five methods tested here provided reliable estimates of alpha and beta diversity.

## Impact of under-sequencing and possible corrections

Finally, to help guide researchers in making decisions about under-sequenced samples, we evaluated how alpha and beta diversity estimates were impacted by such samples in a dataset. Specifically, we performed the same computations (assembly with metaSPAdes from quality-controlled reads, generation of a pool of dereplicated population contigs, mapping of quality-controlled reads and computation of normalized count matrices), but we did so with a dataset in which half of the samples were drastically under-sequenced either at 10% (subset_10) or 1% (subset_1) of the original sequencing depth, respectively (Table S1, Fig. S2).

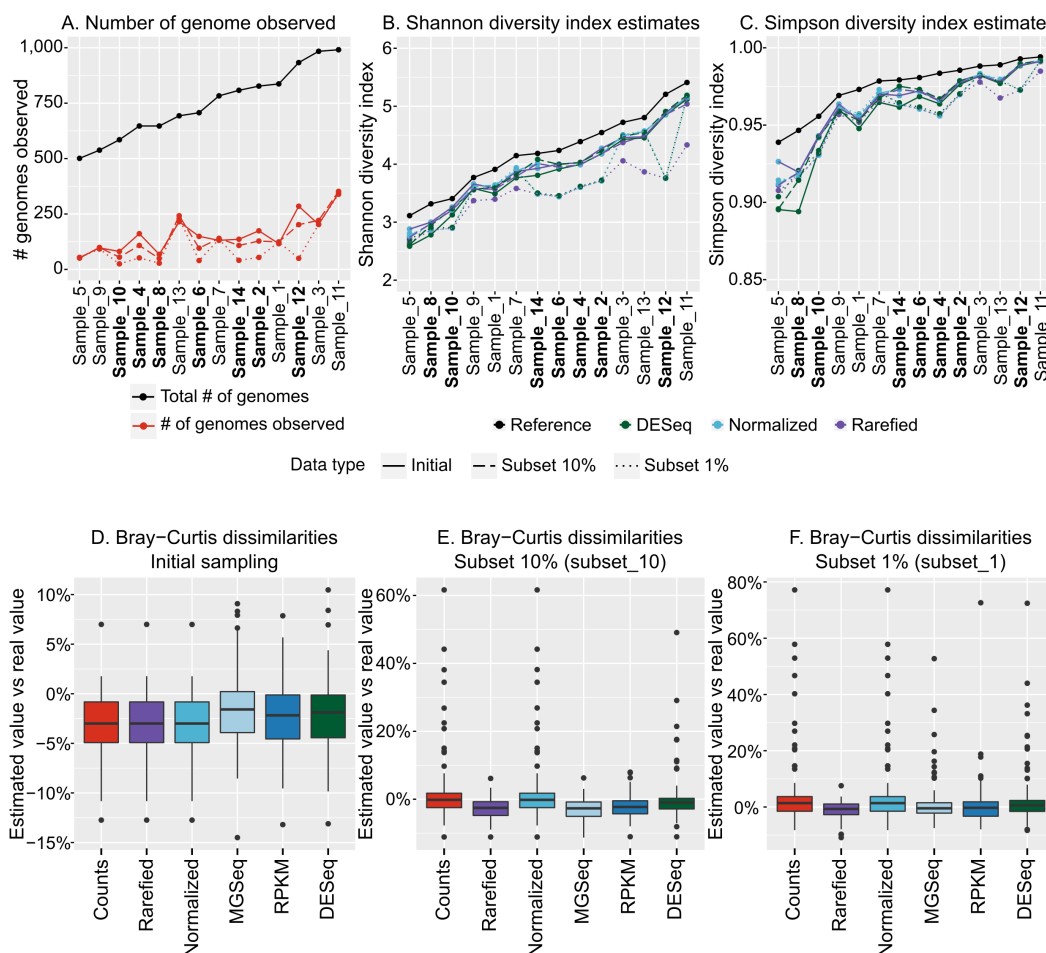

**Figure 4 Estimation of alpha and beta diversity from virome-derived viral populations.** To evaluate the impact of varying sequencing depth, six viromes (highlighted in bold in A–C), were sub-sampled at 10% (long dash) or 1% (short dash) of the original read number ("Initial" corresponds to the assemblies presented in Figs. 1–3, for which all viromes had the same initial number of reads). A. Number of genomes observed from the read mapping to viral populations. The actual number of genomes in the initial simulated community is indicated with black dots, while estimated based on viromes are colored in red. B. Comparison of Shannon diversity index from the true community composition (black dots) and estimated from the viromes (colored dots). The different estimations are based on 3 different normalization methods: counts divided by the total number of reads sequenced in the virome and the contig size ("Normalized"), counts after rarefying all viromes to the smallest dataset and normalized by contig size ("Rarefied"), and counts normalized via DESeq ("DESeq"). (C) Comparison of Simpson diversity index from the true community composition and estimated from the viromes (color codes are the same as in B). (D) Distribution of differences in Bray–Curtis dissimilarities between samples calculated from true community composition and the same dissimilarities estimated from the viromes analysis. The different normalization methods (*x*-axis) are as follows: counts divided by genome size ("Counts"), counts rarefied to the smallest dataset and normalized by contig size ("Rarefied"), counts divided by the total number of reads sequenced in the library and the contig size ("Normalized"), counts normalized by metagenomeSeq ("MGSeq"), EdgeR ("RPKM"), and DESeq ("DESeq"). (E) Distribution of differences in Bray–Curtis dissimilarities between samples calculated from true community composition and the same dissimilarities estimated from virome analysis, including 6 samples sequenced at 10%. Methods are similar as in (D). (F) Distribution of differences in Bray–Curtis dissimilarities between samples calculated from true community composition and from virome analysis, including 6 samples sequenced at 1%. Methods are similar as in (D).

Not surprisingly, under-sequenced samples resulted in fewer genomes detected ($t$-test, $p$-value $< 1e-05$, Fig. 4A). Using the same five normalization methods to account for these differences in sequencing depth, we found that the diversity estimations were impacted. The subset_10 samples resulted in Shannon and Simpson estimations that were close (within 16%) to the initial estimates, but the diversity estimates in the subset_1 samples varied as much as 30% (Figs. 4B & 4C). Hence, although the different normalization methods tested here helped to compensate for some degree of under-sequencing, none was able to recover the correct values of alpha diversity when sequencing depth was highly variable and/or when some samples were significantly under-sequenced.

Similarly, beta diversity patterns (evaluated as pairwise Bray–Curtis dissimilarities) were not estimated as accurately with the under-sequenced samples than with the initial samples: dissimilarities estimated from subset_10 samples varied as much as 61% compared with the true dissimilarities (mean: 5.9%), and the ones estimated from subset_1 samples varied as much as 77% (mean: 4.4%; Figs. 4E & 4F). Rarefaction and MGSeq were the two normalization methods most efficient at limiting these biases, as they led to maximum variations of 11.5% and 11.3% for subset_10, and 10.9% and 52.7% for subset_1, respectively. Moreover, even with the subset_1 samples, the results of an NMDS based on these normalized count matrices were still strongly correlated with the results of an NMDS based on true relative abundances (Fig. S8, $r^2 > 0.9$ for all normalization methods but "rarefied", for which the positions of two groups are switched leading to a lower $r^2$ of 0.64). Hence, beta diversity trends can be recovered even when sequencing depth was highly variable.

Although not formally evaluated through *in silico* benchmarks, it is very likely that microbial metagenomes with highly uneven sequencing depth would be subjected to similar biases, and the tools tested here would be expected to perform comparably on viral and microbial metagenomes, since the input data (i.e., coverage matrix) is essentially identical. Hence, the information and guidelines provided here can in all likelihood be considered relevant for microbial metagenomes as well.

## Current limitations of the sample-to-ecological-inference pipeline

Overall, these benchmarks confirmed that virome-derived abundance matrices can be used in ecological studies, with two main caveats. First, absolute viral richness will likely be under-estimated, because the assembly will only yield large contigs for abundant viral genotypes without evenly distributed and/or closely related strains. Hence, absolute values of richness and diversity should be interpreted with care, although once normalized, sample comparisons of these richness and diversity metrics are generally robust to differences in community complexity and sequencing depth. Second, because this approach relies on coverage as a proxy for relative abundance, only quantitative (or near-quantitative) datasets can be used as input (*Duhaime et al., 2012*). Notably, protocols to generate these quantitative viromes are currently available only for dsDNA and/or ssDNA viruses (*Duhaime et al., 2012*; *Roux et al., 2016*), and still remain to be developed for their RNA counterparts, although these RNA viruses might represent up to half of the viral particles in some environments (*Steward et al., 2013*). Thus, when interpreting viromics-based

ecological studies, it is important to remember and clearly state that these reflect only the sub-part of viral communities with (ds)DNA genomes.

## CONCLUSIONS

Our comparative analysis of 14 simulated viromes showed that the genome-assembly-to-ecological-inference viromics pipeline can efficiently and robustly identify abundant viruses and recover trends in alpha and beta diversity. As viromics becomes routine in viral ecology, the approaches underlined here (both the tools and thresholds used) offer an initial set of "best practices" for data analysis.

Moving forward, increased library size and number associated with improved genome recovery from metagenomes will undoubtedly lead to an unprecedented catalog of uncultivated viral genomes (e.g., 125,000 released in a single study; *Paez-Espino et al., 2016*). These will be complemented by viral genomes obtained from other methods, such as single-virus sequencing, which can access less dominant viruses and those with high strain heterogeneity (*Martinez-Hernandez et al., 2017*). As standards emerge, such uncultivated viral genomes will migrate toward specifically-designed databases (e.g., IMG/VR, *Paez-Espino et al., 2016*), and viral ecological studies will be greatly improved by these centralized reference genome data. Beyond improved references (which will also need to include uncultivated RNA viruses), viromics will need to advance from relative abundance estimations to absolute quantification of viral populations, likely coupled with "ground-truthing" provided by quantitative, lineage-specific molecular methods such as phageFISH, polonies, microarrays, or microfluidic PCR (*Tadmor et al., 2011*; *Allers et al., 2013*; *Martínez-García et al., 2014*). Once in-hand, such approaches should enable researchers to address long-standing questions in the viral ecology field, and more fully bring viruses into predictive ecological models across Earth's ecosystems.

**List of abbreviations**

| | |
|---|---|
| **ANI** | Average Nucleotide Identity |
| **ANOVA** | ANalysis Of Variance |
| **FDR** | False Discovery Rate |
| **NMDS** | Non-metric MultiDimensional Scaling |
| **OTU** | Operational Taxonomic Unit |
| **QC** | Quality-controlled (for reads) |

## ACKNOWLEDGEMENTS

High performance computing resources were provided by the Ohio Supercomputer Center, and the National Energy Research Scientific Computing Center supported by the Office of Science of the US Department of Energy.

### Funding

Matthew B. Sullivan and Simon Roux were supported by grants from the Gordon and Betty Moore Foundation (GBMF #3790) and NSF Biological Oceanography (OCE-1536989) awarded to Matthew B. Sullivan. Joanne B. Emerson was supported by the US Department of Energy, Office of Science, Office of Biological and Environmental Research, under the Genomic Science program (Awards DE-SC0010580 and DE-SC0016440). The work conducted by the US Department of Energy Joint Genome Institute, a DOE Office of Science User Facility, is supported under Contract No. DE-AC02-05CH11231. The funders had no role in study design, data collection and analysis, decision to publish, or preparation of the manuscript.

### Grant Disclosures

The following grant information was disclosed by the authors:
Gordon and Betty Moore Foundation: GBMF #3790.
NSF Biological Oceanography: OCE-1536989.
US Department of Energy, Office of Science, Office of Biological and Environmental Research: DE-SC0010580, DE-SC0016440.
US Department of Energy Joint Genome Institute: DE-AC02-05CH11231.

### Competing Interests

The authors declare there are no competing interests.

### Author Contributions

- Simon Roux conceived and designed the experiments, performed the experiments, analyzed the data, wrote the paper, prepared figures and/or tables, reviewed drafts of the paper.
- Joanne B. Emerson, Emiley A. Eloe-Fadrosh and Matthew B. Sullivan conceived and designed the experiments, analyzed the data, wrote the paper, prepared figures and/or tables, reviewed drafts of the paper.

### Data Availability

The scripts used in this study are available at https://bitbucket.org/MAVERICLab/benchmarking_viromics.

The datasets generated for this study are available at http://datacommons.cyverse.org/browse/iplant/home/shared/iVirus/Virome_pipeline_benchmark.

### Supplemental Information

Supplemental information for this article can be found online at http://dx.doi.org/10.7717/peerj.3817#supplemental-information.

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
