# Peer review of "Benchmarking viromics: an in silico evaluation of metagenome-enabled estimates of viral community composition and diversity"

_PeerJ, doi:10.7717/peerj.3817_

## Round 0.1 · original submission · Minor Revisions

Reviewer #1 has quite detailed requests to be made regarding your manuscript, in particular points 1 and 2 in the Validity of Findings. I thank that both of these issues could be addressed in a timely manner, but please also address the general comment from Reviewer #2 (regarding definitions of "community" in the Introduction) and the additional comments from Reviewer #1 regarding clarity.

Reviewer 1 ·

Basic reporting

Minor comments:

1. The first figure reference in the text is to Fig. S1F (line 127). What about Figs. S1A-E? Figure references in the text should occur order. Please fix this.

2. It would be helpful to state in the methods section, the read length used in the simulation of metagenomes. I realize that it is stated parenthetically in the results section (line 277), but this is an important parameter that should be stated in the methods section.

3. In the methods section, line 179, it should be explicitly stated what the term “MEGAHIT-QC reads assembly” means. Do you mean assembly using MEGAHIT on QC reads? Please clarify

4. The one exception to clarity (see general comments section) in results presentation is the section on “Population identification and quantification” (Figs. 3A,D), where I found that the structure and language used made it difficult to easily understand their evaluation criteria. The sentences at the end of the paragraph describing the mapping threshold was clear (lines 404-406), but there was another threshold that was applied in their analysis, only considering contigs with >= 1X coverage, that is only explicitly mentioned if you look carefully at the axis label of Figures 3A,D and no where else. This important criterion should be stated in the main text and the figure legend. Can you also clarify what “among genomes covered >=1X” means? I assume it means that you only considered genomes that were sampled at >=1X in the read simulation step, not that the assembled contigs for that genome were detected at >=1X. This should be stated more clearly.

5. Line 474. Please make sure that the terms used for under-sequenced datasets are consistent in the text and figures and tables. Specifically, the terms “subset_10” and “subset_1” are used in the text, but these terms do not appear in Figure S2 and Table S1.

6. Axes labels on Figs. 1 & 4 are a bit too small to easily read, and they are definitely too small for Figs. 2 & 3, especially Fig. 3. Please fix this.

Experimental design

I was curious to know if assembly was attempted on non-QC reads. I have heard some debate among my colleagues that QC'ing reads actually isn't often necessary for assembly. Is this something you investigated in your study but have not shown results for? Can you please comment on this and if it would be a valuable analysis to add to this study?

Validity of the findings

Major comments:

1. The abstract states that “metaSPAdes and MEGAHIT were the most effective at assembling viromes”, but it was not clear to me from the results or discussion from the results why IDBA-UD is excluded from this list. In fact, the Fig. 1 results put them on par and the text states “Together these comparisons suggest that: (i) IDBA-UD, MEGAHIT, and metaSPAdes are currently the best available choices for maximizing assembly of viral contigs from short-read (100 bp) viromes (assembly accuracy discussed below)” (Line 305). Is preference given to MEGAHIT over IDBA-UD because MEGAHIT generates less chimeric contigs. I think if you are going to highlight two assemblers in the abstract, there should be some clear criterion for doing some in the main text.

2. I think there should be more discussion of how their results compare to similar benchmarking studies that have been done for prokaryotic metagenomes. I recognize that the purpose of the paper is to focus on working with viral metagenomes, but many of the issues (i.e. sensitivities of assembly to the particular assembler used, coverage, evenness of samples, strain variation) I expect should be very similar to that for prokaryotic assembly that has been extensively tested previously. It think it would be valuable for the reader to know if the results in this study are similar to what has been found for bacterial and archaeal metagenome assembly. For example, do IDBA-UD, MEGAHIT, and metaSPAdes perform better than MetaVelvet and Omega in genome recovery? How about for chimera results and strain heterogeneity tests? Are there similar studies that have looked at read mapping thresholds and diversity estimates? I would expect that there would be similar findings for the viral assembly results here and prokaryotic assembly benchmarking studies. If there are notable differences, these should be discussed briefly as to why.

If some of their benchmarking studies in this study have no comparable parallel in the literature for benchmarking prokaryotic metagenome analyses, then I think the authors could highlight how their new benchmarking analyses for viromes could be informative to bacterial and archaeal metagenome assembly as well.

I imagine that these discussion points could be easily incorporated as a few sentences at the end of each evaluation section.

Additional comments

Roux et al. provide a thorough analysis of various tools for assembling and basic characterization of resulting contigs (alpha and beta diversity). This study will provide a very useful reference for scientists doing work on viromes, and it parallels various studies that have benchmarked assembly and diversity measurement tools for bacterial and archaeal metagenomes. A strength of this study is that it provides some practical guidelines for assembly and analysis of virome data.

The authors have framed their findings, in most instances, in very concise and easy to understand writing and figures. See the one exception under minor comments.

·

Basic reporting

The article is clear, unambiguous, and easy to read. The work is well-motivated; the experimental design is clearly justified and presented; results and conclusions stated explicitly, together with appropriate caveats.

My own background is in bioinformatics, rather in virology or viral ecology, so I can't assess the realism of the models. etc. My interest, though, is in the design of methods and the possible impact that sequencing and assembly issues might have on the analysis of data. For this I found that the manuscript was especially useful.

Experimental design

As I said, I'm really not qualified to assess whether this is a 'realistic' replicate of viral communities. However the design is very clearly laid out and justified. Sufficient details are provided to repeat it as necessary, and, to my knowledge, the authors have covered the major analysis pathways in practice (and several known sources of potential error).

Validity of the findings

Covered above.

Additional comments

One minor question that could be easily dealt with in the introduction is that I'm unsure exactly what the authors are thinking of when they speak of a 'viral comunity'. Should I be thinking of the viral load of an individual? Or viruses in a community of of bacteria? of eukaroytes? Or maybe of the 'community' of influenza viruses infecting humans in one season. Each of these will be modelled differently.

---

## Round 0.2 · accepted · Accept

All issues have been addressed. Please read carefully through the manuscript for any typos and grammatical problems one last time!